# Recent Advancements in Graphene-Based Implantable Electrodes for Neural Recording/Stimulation

**DOI:** 10.3390/s23249911

**Published:** 2023-12-18

**Authors:** Md Eshrat E. Alahi, Mubdiul Islam Rizu, Fahmida Wazed Tina, Zhaoling Huang, Anindya Nag, Nasrin Afsarimanesh

**Affiliations:** 1School of Engineering and Technology, Walailak University, 222 Thaiburi, Thasala District, Nakhon Si Thammarat 80160, Thailand; 2Microsystems Nanotechnologies for Chemical Analysis (MINOS), Universitat Rovira I Virgili, Avinguda Països Catalans, 26—Campus Sescelades, 43007 Tarragona, Spain; mubdiulislam.rizu@urv.cat; 3Creative Innovation in Science and Technology Program, Faculty of Science and Technology, Nakhon Si Thammarat Rajabhat University, Nakhon Si Thammarat 80280, Thailand; fahmida_tina@nstru.ac.th; 4School of Mechanical and Electrical Engineering, Guilin University of Electronic Technology, Guilin 541004, China; zhaoling_huang@guet.edu.cn; 5Faculty of Electrical and Computer Engineering, Technische Universität Dresden, 01062 Dresden, Germany; anindya.nag@tu-dresden.de; 6Centre for Tactile Internet with Human-in-the-Loop (CeTI), Technische Universität Dresden, 01069 Dresden, Germany; 7School of Civil and Mechanical Engineering, Curtin University, Perth, WA 6102, Australia; nasrin.afsarimanesh@curtin.edu.au

**Keywords:** graphene, implantable electrode, neural recording/stimulation, GFET recording

## Abstract

Implantable electrodes represent a groundbreaking advancement in nervous system research, providing a pivotal tool for recording and stimulating human neural activity. This capability is integral for unraveling the intricacies of the nervous system’s functionality and for devising innovative treatments for various neurological disorders. Implantable electrodes offer distinct advantages compared to conventional recording and stimulating neural activity methods. They deliver heightened precision, fewer associated side effects, and the ability to gather data from diverse neural sources. Crucially, the development of implantable electrodes necessitates key attributes: flexibility, stability, and high resolution. Graphene emerges as a highly promising material for fabricating such electrodes due to its exceptional properties. It boasts remarkable flexibility, ensuring seamless integration with the complex and contoured surfaces of neural tissues. Additionally, graphene exhibits low electrical resistance, enabling efficient transmission of neural signals. Its transparency further extends its utility, facilitating compatibility with various imaging techniques and optogenetics. This paper showcases noteworthy endeavors in utilizing graphene in its pure form and as composites to create and deploy implantable devices tailored for neural recordings and stimulations. It underscores the potential for significant advancements in this field. Furthermore, this paper delves into prospective avenues for refining existing graphene-based electrodes, enhancing their suitability for neural recording applications in in vitro and in vivo settings. These future steps promise to revolutionize further our capacity to understand and interact with the neural research landscape.

## 1. Introduction

The implantable microelectrode array has been instrumental in comprehending regular neural processes, studying neurological behaviors, and facilitating bidirectional communication between electronic devices and the central nervous system (CNS) [1]. Conventional clinical procedures involve the insertion of electrodes into the brain, often employing sharp silicon-based variants for critical applications. Unfortunately, this method presents notable drawbacks, including potential inflammation and damage from the implantation process. Minimally invasive techniques have been explored to mitigate these concerns, focusing on implementing flexible microelectrode arrays. Notably, graphene has emerged as a key contributor to recent advancements in this field. A flexible microelectrode array can record the neural signal and stimulate the neurons to understand the neuronal behavior of the cortical circuits. Despite technological advancement and innovation, the implantable microelectrode array has significant challenges and limitations [2,3,4].

The current era of science and technology has witnessed the development of highly efficient sensing devices for various intrinsic human body signals. Among these, the design and advancement of high-quality neural recording and stimulation systems have been particularly impactful [5,6,7]. These signals are pivotal in comprehending the fundamental processes within the neural and neurological domains. In clinical practices, microelectromechanical (MEMS) silicon-based electrodes have been employed for intracranial applications [7,8,9]. However, this approach has drawbacks, including side effects like inflammation and implantation damage. Consequently, there has been a paradigm shift toward considering flexible sensing prototypes and electrodes as a cornerstone for the past two decades [10,11]. These electrodes have been fabricated using various materials and techniques, resulting in electrodes with optimized electromechanical parameters for specific applications. One popular category among these flexible electrodes, particularly in biomedical contexts, is flexible microelectrode arrays due to their non-invasive nature [12,13,14]. Scientists have made significant progress in enhancing electrode performance, addressing key aspects like long-term signal stability, detection sensitivity, multifunctionalization, and in vivo biocompatibility [3,15,16,17,18,19]. Conventional brain electrodes have historically leaned toward the use of noble metals such as Ag [20], Au [21], and Pt [22] for their stability and ease of production. Their limited electron transfer capabilities have constrained neural signal detection sensitivity. As a result, alternative nanomaterials have been explored for electrode development, emphasizing specific characteristics such as non-invasiveness, robust interfacing, high charge transfer capacity, and low output impedance [23]. While conventional electrodes made from pure metals or alloys are adequate to a certain extent, modifications involving compounds like iridium oxide, titanium nitride, PEDOT, and carbon nanotubes have been pursued to enhance charge injection capacity for improved simulation capabilities [24]. However, the poor adhesion of these modified electrodes has been a significant challenge, leading to delamination during real-time neural recordings [16].

Therefore, these electrodes have been formed using various nanomaterials to form enhanced electrodes with high sensitivity and selectivity. Some of the common materials used to create flexible electrodes are carbon-based elements like carbon nanotubes (CNTs) [25,26,27,28], graphene [29,30,31,32], graphite [33,34,35], and other metallic elements like copper [36,37,38], platinum [39,40,41], gold [42,43,44], and silver [45,46,47]. These elements effectively form excellent flexible electrodes due to their enhanced electrical, mechanical, and thermal characteristics. Enhanced graphene is an emergent two-dimensional nanomaterial boasting remarkable traits including high transparency (497.3%), low electronic resistivity (1.00 × 10^−8^ Ω·m), and exceptional electron mobility (105 cm^2^·V^−1^·s^−1^) [48,49,50,51,52,53,54,55,56]. It forms a stable electrode–neural tissue interface, closely integrating with brain tissue, thus holding immense promise as a next-generation neuronal electrode [48,49,50,51,52]. The electrical signal transfer is much more sensitive thanks to graphene’s exceptional electrical and optical properties. Due to this, optical imaging and optogenetic stimulation techniques can be used directly on brain tissue, enhancing the spatial and temporal resolution of neural activity detection [53,54]. Moreover, graphene-related electrodes’ tensile flexibility has the potential to facilitate wound healing, diminish scar damage, and reduce tissue inflammation, making them highly viable for long-term neural signaling activity monitoring [55,56]. Researchers have extensively utilized graphene and other materials to create sensors with exceptional physiochemical properties, making it a crucial element in developing prototypes for neuro-interfacing applications. The two-dimensional (2D) hexagonal carbon lattices are linked by covalent solid in-plane σ-σ bonds, supplemented by extra π-π bonds within the electronic orbitals aligned along the vertical plane of the graphene [57]. This leads to electrons’ delocalization, which is vital for electrochemical sensing applications. Due to the zero bandgap between the valence and conduction bands of graphene, a solid ambipolar electric field effect is observed with a high charge carrier mobility of ≈10,000 cm^2^·V^−1^·s^−1^ at room temperature [58]. In addition, the white light absorbance, thermal stability, mechanical strength, and specific surface area are 2.3%, 3000 ≈ 5000 W·m^−1^·K^−1^, ≈1 TPa, and 2630 m^2^·g^−1^, respectively [59]. These superior characteristics of graphene are an ideal candidate for applying strain [60,61,62], electrochemical [63,64,65,66], and electrical [67,68,69] applications. In the realm of electrochemical and electrical applications of graphene for signal detection within the body, establishing direct contact and exposing neurons to graphene-based electrodes proves imperative. This ensures a close adhesion between cell membranes and the interfacing electrodes. This adhesive behavior leads to detecting small signals, in order of microvolts, during the extracellular recordings and tissue stimulation processes [70]. The remarkable electrical conductivity and minimal noise characteristics of graphene present a promising avenue for upscaling the structural dimensions of the electrodes, ranging from single-cell measurements to macro-sized ones, by improving the SNR. These properties also reduce the electrical impedance and improve the charge injection capacity (CIC). The CIC refers to the maximum amount of electric charge that can be delivered or extracted by an electrode interface without causing irreversible chemical reactions or damage to the electrode or surrounding tissue. In the context of neural stimulation or recording, CIC is an important parameter to consider for electrodes used in implantable devices. For neural stimulation, the electrode must deliver a controlled charge to activate neurons without causing harm. On the other hand, in neural recording, the electrode should be capable of detecting minute electrical signals generated by neural activity. The high mechanical and chemical stability is excellent for using graphene to develop flexible sensors, especially for soft biological tissue interfaces (Figure 1). One such application has been highlighted in this paper.

In the landscape of prior review papers, various research teams have skillfully outlined recent advancements in graphene-based electrodes and their applications across neuroscience and technology [72,73,74,75,76]. Researchers have demonstrated the potential of graphene in fields such as nanomedicine [77,78], biosensors [79,80], and flexible electronics [81,82]; these reviews have collectively showcased its outstanding electrical properties, particularly for high-fidelity neural signal recording [83]. Building upon this foundation, our review sets itself apart by uniquely focusing on the intricate interface between neural tissue and graphene, highlighting the distinctive attributes that make graphene an exceptional candidate for electrophysiological recording and stimulation. Unlike prior reviews, our analysis delves deeply into specific sections covering neural tissue interface enhancement, implantable electrodes, equivalent circuit modeling, and more, providing a comprehensive exploration of the potentialities of graphene in neural interfaces. Additionally, we offer a forward-looking perspective on the future trajectories of this evolving field, aiming to contribute novel insights to the discourse on graphene-based neural interfaces.

## 2. Neural Tissue Interface Enhancement with Graphene

The intricate interplay of neural cells within brain tissue generates essential neuroelectric signals through electrophysiological activity. This process involves the passage of potassium and sodium ions via ion channels, thereby influencing the extracellular potential [39]. Neural electrodes have revolutionized our ability to record extracellular potential changes, offering critical insights into neural activity and underlying pathological mechanisms. These electrodes facilitate the non-invasive examination of neural activity in vitro, providing valuable data for in vivo studies without invasive procedures on living organisms.

The implantable electrode interface links brain–computer interface (BCI) devices and neurons within the central nervous system. This interface is instrumental in advancing our understanding of various neurological processes and restoring function in disorders like epilepsy, paralysis, Alzheimer’s disease, and motor dysfunction due to limb loss. Both implantable electrodes and BCI devices play an integral role as neural interfaces, enabling neural stimulation and recording while maintaining high signal quality and minimizing noise arising from individual neurons—commonly referred to as action potentials [39]. Neural signals are categorized into electroencephalography (EEG) [55], electrocorticogram (ECoG) [56,57], local field potentials (LFPs) [58], and action potentials (APs) [59] based on the location of the recording sites.

The choice of implantable electrode positioning significantly affects the quality of neural signals captured using various neural recording technologies. Electroencephalography (EEG) is a non-invasive technique widely employed for observing sleep patterns and understanding brain activity, particularly for conditions like seizure treatment [60,61,62,63]. However, EEG signals can be susceptible to interference from local field potentials (LFPs) and have limitations in capturing signals from specific brain regions due to low transfer rates, typically ranging from 5 to 25 bits per second (bps) [64,65]. Moreover, the densely packed nature of brain tissues, coupled with intervening layers like skin and the skull, obstructs EEG signals, compromising their spatiotemporal resolution [3].

Electrocorticography (ECoG), on the other hand, offers advantages for minimally invasive neural recording purposes, surpassing EEG’s limitations. ECoG significantly reduces noise levels and enables the high-frequency and accurate recording of neural signals. When placed on the cortex, implantable ECoG electrodes mitigate interference from neighboring tissues, resulting in higher quality signals [66]. Nonetheless, ECoG has difficulties capturing individual neural signals from neurons and superficial regions. To address the need for precise, unique neural signal recording across specific cortex areas, implantable local field potentials (LFPs) are utilized, primarily from deeper brain regions. LFPs allow the extraction of local neural activities from precise sites, including action potentials (APs) and membrane potential fluctuations, thus providing valuable insights into specific brain areas [67].

The selection of an appropriate neural signal recording method hinges on the application’s focal point, electrode design, material attributes, and the chosen implantation site. Recent trends highlight substantial strides in implantable electrode technologies, particularly in surpassing the capabilities of EEG and ECoG by focusing on single-neuron activities. Implantable electrodes, coupled with neural devices, hold promise in controlling epileptogenic regions and addressing Parkinson’s disease through targeted neural stimulation, offering more sophisticated neural interfacing capabilities than EEG and ECoG devices [66,67]. As research endeavors progress, innovations in implantable device technologies encompass enhanced spatial resolution, augmented recording sites, and multifunctionality tailored to various neural activities. Integrating simulations, materials science, mechanical design, and electronic engineering accelerates chronic in vivo neural recordings and stimulations facilitated by implantable electrodes alongside BCI devices [70,71,72,73,74,75,76]. A central objective entails the development of fully miniaturized, biointegrated, flexible, and wireless BCI platforms, ensuring mechanical compatibility and seamless integration with neural tissues [77,78].

Establishing stable graphene–neuron interfaces (refer to Figure 2) holds significant implications. Innovative techniques, like a graphene liquid-gate transistor packaging method, help alleviate bending stress, enhancing interface stability [59]. However, challenges surrounding graphene transfer between substrates are addressed through ultraviolet ozone treatment, effectively reducing contact noise and improving electrical performance [60]. These findings underscore graphene’s promise in establishing stable interfaces with neurons, thus augmenting implantable electrodes’ capabilities for precisely recording neural signals.

## 3. Potentiality of Graphene for Implantable Electrodes

Traditional neural electrodes primarily employ metal-based materials, a common choice in biomedical applications. However, there is a growing interest in exploring alternative materials for enhanced performance and biocompatibility [38,84,85,86]. However, they often grapple with high impedance, negatively impacting their recording sensitivity. Moreover, when in contact with tissue, these electrodes can lead to uneven electron transfer, potentially causing discomfort and tissue damage at the interface with brain tissue. Despite efforts to integrate materials like silicon and flexible polymers into electrode production, impedance continues to pose a considerable hurdle [87]. Graphene’s exceptional biocompatibility and electrical properties make it ideal for interfacing with neural tissues. Graphene’s capacity to precisely record neuro-electrophysiological activity with high temporal resolution and stable regulation presents unprecedented advantages for microengineering applications. Notably, graphene’s distinct edge over conventional metal electrodes lies in its transferability onto transparent substrates, thereby enabling the creation of transparent neural electrode arrays. Moreover, it is essential to highlight that doped graphene exhibits higher transmittance than ITO and ultrathin metals. This characteristic eliminates artifacts when high-intensity light is directly applied to an electrode site, making doped graphene particularly advantageous. Such enhanced transmittance facilitates the successful implementation of various optical techniques, including optogenetic stimulation, optical coherence tomography (OCT), and fluorescence imaging, beneath the graphene electrode sites [88]. Table 1 presents a comparative analysis of flexible electrode materials, highlighting the distinct advantages of graphene over other counterparts. Graphene, with its exceptional electrical conductivity of 243.5 ± 15.9 kΩ (~200 µm diameter) and extreme flexibility (~1 TPa), emerges as a superior choice. Its high transparency and excellent biocompatibility further enhance its desirability for various applications. In contrast, other materials like PEDOT, PT, PPY, PANI, carbon nanofiber (CNF), glassy carbon, and diamond exhibit varying levels of conductivity, flexibility, transparency, and biocompatibility, with graphene consistently demonstrating superior performance in multiple key aspects.

A significant breakthrough was achieved in 2009 by Kim, who led the way in producing graphene films and was instrumental in advancing the chemical vapor deposition method on thin nickel layers [100]. These films demonstrated meager resistance (below 280 ohms per square) and optical transparency exceeding 80% [100]. Furthermore, graphene films on SiO_2_ substrates exhibited impressive electron mobility of 3700 cm^2^ V^−1^ s^−1^ [100]. These attributes substantially enhanced the detection of neuronal signals. This innovative approach not only facilitated electrophysiological signal recording but also upheld the quality of imaging data. Electrodes with increased sensitivity for electrophysiological recording are essential for diagnosing neurological illnesses. Traditional opaque brain electrode arrays frequently have poor spatial resolution and sensitivity, which causes problems like image artifacts and data loss. As a result, graphene, with its exceptional properties, including transparency, flexibility, and controlled electrical conductivity, emerges as a highly promising material for developing advanced brain electrodes.

## 4. Characteristics of Graphene-Based Implantable Electrodes

Implantable electrodes made from graphene represent a revolutionary leap forward in the field of neural engineering. These electrodes exhibit exceptional properties that make them highly attractive for various applications within the field and are crafted from a single layer of carbon atoms meticulously arranged in a two-dimensional honeycomb lattice. The subsequent sections will delve into some of the significant features of these electrodes.

### 4.1. Flexible Electrodes

The insertion of rigid-surfaced electrodes, including materials like gold, platinum, iridium, stainless steel, and tungsten, for clinical and biological research purposes can inadvertently damage neurons. This phenomenon occurs as these hard electrodes are implanted, leading to the destruction of surrounding neural tissue. Additionally, widely used silicon-based implantable electrodes, exemplified by the Utah electrode array [101] or Michigan electrode, have become staples in neuroscience research. However, their inherent rigidity poses limitations for long-term neural recording and stimulation due to the potential for tissue damage and reduced compatibility. As a result, alternative approaches and materials are sought to address these drawbacks and enable safer and more effective long-term applications in neuroscientific investigations.

In this context, graphene emerges as a promising candidate. With its distinctive hexagonal honeycomb structure of carbon atoms, graphene exhibits remarkable electrical, mechanical, and chemical characteristics. Notably, its inherent flexibility allows for seamless integration with surrounding tissues, mitigating the potential harm to nerve tissues during implantation. This unique feature positions graphene as a viable replacement for conventional materials like silicon and metal in the development of innovative neural interfaces [102,103]. The porous graphene electrode, in particular, stands out for its expansive low impedance, specific surface area, and robust charge injection capacity, collectively elevating the standard of cortical recording and stimulation quality [104].

Kuzum and colleagues played a pivotal role in pioneering the fabrication of three-dimensional porous graphene foam. This innovative material was crafted through a direct etching process on a polyimide substrate, utilizing laser pyrolysis as the key technique [105]. They incorporated Cr/Au metal leads and contact pads into the structure. As an encapsulation layer, they employed negative photoresist SU-8. This procedure resulted in a flexible graphene neural electrode array renowned for its heightened porosity and surface irregularities (Figure 3). Remarkably, the impedance of this array was nearly a hundredfold lower than that of gold electrodes with comparable dimensions. Additionally, a chemical doping process involving nitric acid treatment was employed to diminish impedance further, concurrently augmenting the charge injection limit (CIL) from 2 to 3.1 mC·cm^−2^. The heightened CIL bears substantial implications for electrode performance, signifying the electrode’s ability to deliver charge efficiently while ensuring safety thresholds are not exceeded for both the surrounding tissue and the electrodes themselves. The achieved CIL value demonstrates exceptional suitability for a broad spectrum of applications, outperforming materials such as Ir_x_O, carbon nanotubes, PEDOT, Ta_2_O_5_, and titanium nitride. Following this success, the electrodes were carefully positioned on the surface of the rat sensory cortex to record sensory-evoked potentials. Notably, applying stimulation to the motor cortex using these electrodes led to a distinct flexion in both ankle and knee joints. Particularly in the fields of electrical microstimulation and the mapping of spatial–temporal cortical dynamics, this groundbreaking study provides a potent instrument.

Garrett and his team employed a thorough wet spinning technique to create fibers from graphene oxide. Following this, the fibers underwent annealing at 220 °C, forming liquid crystal graphene oxide (LCGO) fibers [106]. To provide insulation, a protective layer of Parylene C was applied. The fiber endings underwent precise laser ablation, yielding a neural electrode with significantly increased charge injection capacity. This process increased surface roughness and nanoporosity creation, as illustrated in Figure 4. The study team used this state-of-the-art electrode to stimulate ganglion cells in a detached rat retina in an in vitro setting. They also recorded a thorough whole-cell patch clamp simultaneously. The electrode’s surface was carefully coated with a water-soluble sucrose coating to create micron-scale needles. The cat’s visual brain was then implanted with this modified flexible electrode. The sucrose layer was then removed, making it easier to monitor brain activity. Creating a self-supporting, flexible shank that effortlessly combines with the electrode was a significant development in this work. With this invention, there was no longer a need for complex material interfaces or the time-consuming procedure of connecting a larger electrode to a smaller wire.

Graphene’s versatility extends to its integration with various materials for fabricating neural electrodes, harnessing the strengths of multiple substances. In a pioneering study conducted by Jang and colleagues, they introduced a neural probe with a recording site crafted from a composite material amalgamating poly(3,4-ethylenedioxythiophene) (PEDOT), gold (Au), and zinc oxide (ZnO) nanowires. Complementing this, a lead wire constructed from a combination of gold (Au) and graphene was also integrated into the probe [107]. The integration of graphene with various materials in neural electrode fabrication shows its remarkable versatility. Including ZnO nanowires with a PEDOT coating significantly increased the electrode’s surface area and charge storage capacity, reducing impedance. The Au–graphene lead exhibited remarkable flexibility and conductivity, showcasing the reinforcing effect of graphene on the electrode’s resilience to bending. This combination holds great potential for enhanced neural electrodes.

### 4.2. Transparent Electrodes

Transparent neural interfaces play a pivotal role in minimizing light-induced artifacts and ensuring conductivity for the precise measurement of electrical signals [108,109]. Achieving optimal electrochemical impedance between the electrode sensing site and tissue is crucial for the effective transmission of neural signals. Additionally, minimizing trace resistance between the sensing site and the percutaneous connector is essential for obtaining high-quality electrophysiological signals. When designing transparent electrode arrays, material selection becomes paramount. Carbon or polymer-based electrodes, while offering transparency, often exhibit unfavorable electrical properties compared to metal wiring in terms of trace resistance, especially when considering the same interconnect line width and length. In the case of carbon-based neural electrode arrays, conductive metals are frequently introduced to enhance the electron path intuitively, albeit at the expense of interconnect transparency. On the other hand, polymer electrode arrays can mitigate trace resistance by modifying the molecular structure through specialized doping or post-treatment, maintaining transparency throughout the device, including the conducting path [110,111].

Prioritizing both high transparency and conductivity is crucial when selecting materials for multimodal device design. Recently, graphene has emerged as a material of significant interest for transparent neural interfaces. Beyond its remarkable electrical conductivity, graphene boasts excellent transparency, attributed to its unique two-dimensional honeycomb structure. Graphene exhibits exceptional intrinsic optical transparency, surpassing 90%, across ultraviolet (UV) and infrared (IR) light spectrums [112,113]. This remarkable transparency is particularly significant for transparent neural interfaces, as it ensures high visible light transmittance, thereby minimizing optical blocking. In the context of transparent neural interfaces, maintaining high visible light transmittance is crucial for optimal performance. Moreover, the high optical transmittance of neural interface materials, extending to both UV and IR light, holds significant advantages for applications such as optogenetic stimulation and photo-induced imaging [114]. These properties enhance the versatility of transparent neural interfaces, enabling precise stimulation and imaging techniques.

Williams et al. [95] introduced a carbon-layered electrode array (CLEAR) employing four graphene layers on a Parylene C substrate, featuring 16 electrode locations with exceptional transparency surpassing 90% across UV to IR spectra. Adjusting optical power effectively mitigated artifacts caused by illumination. Positioned precisely in the cerebral cortex of a Thy1:ChR2 transgenic mouse, the CLEAR device optimally responded to 473 nm blue light, seamlessly capturing neuroelectrical signals. It enabled fluorescent imaging and optical coherence tomography of cortical blood vessels at the electrode site, owing to graphene’s broad light transmission spectrum. The transparent electrode (refer to Figure 5) facilitated the unobstructed visualization of underlying tissue, allowing for detailed imaging [88].

In a subsequent investigation, transparent microelectrode arrays fabricated from graphene were employed for micro-electrocorticography (μECoG) studies. This innovative approach enabled the concurrent application of neuroelectrical stimulation and the imaging of neural activity within the cortex of transgenic mice featuring the GCaMP6f indicator [115]. The exceptional light transmittance of graphene facilitated the visualization of neural activity, elicited by electrical stimulation using fluorescent calcium imaging. Remarkably, the graphene electrode exhibited an impressive charge injection limit (CIL) ranging from 116.07 to 174.10 μC·cm^−2^. Furthermore, it was observed that cathodic stimulation elicited a more robust neural response compared to anodic stimulation, affirming a more efficient charge transfer to the brain. The extraordinary light permeability of graphene enabled the observation of neural responses triggered by electrical stimulation via fluorescent calcium imaging.

In their investigation, Kuzum et al. used transparent, bendable graphene electrode arrays to record electrophysiology while simultaneously photographing optical signals [59]. The flexible polyimide substrate, p-type doped graphene site, and SU-8 encapsulation of the doped graphene electrode allowed it to exhibit exceptional properties like low impedance and high charge storage. This unique design allowed for optical artifact-free simultaneous calcium ion imaging and electrophysiological recordings of slices of hippocampus tissue. Furthermore, transparent graphene electrodes demonstrated proficiency in detecting high-frequency electrical activity. This aspect offered high spatial resolution, albeit with lower temporal resolution, providing a valuable complement to calcium imaging. The corrosion of the Ag electrode was notably inhibited by encasing it with graphene. After six months of immersion in a phosphate buffer, it was observed that the graphene-coated Au electrode had been effectively protected. The Raman spectrum post-immersion revealed distinct graphene peaks, confirming that transparent electrodes were created by graphene with minimal interference from noise. Furthermore, it was confirmed that graphene served as a layer that protected metal microelectrodes from corrosion, assuring their long-term stability.

An electrochemical bubbling technique for transferring graphene onto a 50 μm thick polyethene terephthalate substrate was introduced by Thunemann et al. [116]. The resulting graphene sheet underwent rigorous surface cleaning before being shaped into electrode locations to avoid crack development and the lingering presence of organic materials. Developing a transparent 16-channel SU-8 encapsulation-coated graphene microelectrode array was made possible. The impedance was less than 1.5 M at 1 kHz, and it could endure continuous bend (up to 20 times) at a radius of curvature of 5 mm, which falls well below the mouse cortex’s normal bending range, without experiencing any failures. Subsequently, the electrode was situated atop the mouse’s primary somatosensory cortex to facilitate two-photon imaging of interneurons and blood vessels, reaching depths up to 1200 μm. Notably, the adaptability of the electrode was showcased as it permitted the activation of both the local field potential (LFP) and calcium ions in the opposite cheek area with a single pulse. This encompassed the synchronized recording of ion transient signals, capturing LFP signals under optogenetic modulation, performing two-photon imaging of arteriole expansion, conducting simultaneous hemodynamic optical imaging, and registering neuroelectric activity under cheek stimulation.

In electroretinography (ERG) studies, an application has been found for transparent electrodes. An innovation by Duan et al. led to the development of flexible and transparent graphene contact lens electrodes (GRACEs) (depicted in Figure 6a) [117]. These electrodes demonstrated exceptional light transmittance across a wide spectral range, establishing a snug and seamless interface with the cornea. Importantly, no observable harm to the cornea was noted during conventional ERG recording. High-fidelity recordings of various ERG signals were enabled by this electrode design. In the domain of full-field ERG, corneal potential amplitudes recorded by GREACEs were found to exceed those obtained with commercial ERG-Jet electrodes. Furthermore, the adeptness of these electrodes in capturing multifocal ERG signals (as illustrated in Figure 6b) was attributed to the preservation of the eye’s refractive power by the conformal interface. Furthermore, a multilocation see-through graphene electrode array (Figure 6c,d) was utilized to differentiate spatially resolved ERG reactions. The research noted that the magnitude of the ERG signal was most prominent at the cornea’s midpoint, progressively diminishing in the temporal and nasal regions.

Duygu Kuzum et al. created the transparent, flexible graphene electrode array with simultaneous electrophysiology and optical neuroimaging in 2014 [59]. The previous gold pattern was transferred onto a polyimide substrate with CVD graphene produced on Cu. The graphene was then patterned using plasma etching. The entire electrode was insulated using SU-8, except for the graphene spots. Nitric acid was then used to dope the electrodes, lowering the sheet resistance of graphene and causing NO_3_^−^ groups to adhere to the material’s surface, producing p-type doping. The doped G electrode’s phase angle in EIS spectroscopy was constant (−50°) over a large frequency range, indicating more complex charge transport than the Au electrode. The doped G electrode had a capacitive characteristic. Large interface capacitance in brain re-recording electrodes helps reduce the electrode noise caused by resistive charge transfer. Due to the doped G electrode’s low charge transfer resistance and high capacitance, electronic noise could be avoided. Due to the doped G electrode’s high charge storage capacity, the charge transfer amount necessary for neurostimulation electrodes may be increased. Dong-Wook Park et al. produced transparent graphene electrodes for optogenetic applications in another study published in 2014 by the same group. Transferred onto the Parylene C substrate, four-layer CVD growth graphene was then designed. Au had patterned the connection pads, initial parts of the tracks, and electrode sites to ensure a robust mechanical connection to the zero insertion force printed circuit board (PBC). However, Au had not patterned how these elements would meet the brain tissue. The doped, four-layer graphene electrode transmitted 90% of the signal with the lowest sheet resistance possible. Doped graphene has a higher transmittance than ITO and ultrathin metals because there are no artifacts when high-intensity light is directly applied to an electrode site, therefore optogenetic stimulation, optical coherence tomography (OCT), and fluorescence imaging can be successfully carried out beneath the graphene electrode sites (see Figure 7) [88].

### 4.3. Hybrid Graphene Electrode

A hybrid graphene electrode combines the remarkable properties of graphene with other materials to create a composite structure, often surpassing the performance of individual components. This concept is especially crucial in applications demanding specific electrical, mechanical, or chemical characteristics. The integration of nanomaterials with graphene electrodes represents a promising frontier in neural activity recording within the field of neurotechnology. Specifically, the application of nanotechnology in neuroscience has led to the development of nanoelectrodes, which possess critical dimensions on the nanometer scale. Analogous to their minute size, a paradigm shift in electrochemical response control is facilitated by these nanoelectrodes, encompassing individual nanoelectrodes, nanoelectrode arrays (NEAs), and nanoelectrode ensembles (NEEs). Unlike conventional electrodes with millimeter diameters, nanoscale electrodes facilitate rapid mass transport through radial diffusion, expediting electrochemical reactions by removing the limitations of mass transport.

Consequently, this progress harbors significant promise for a range of neural interface applications, notably in amplifying sensitivity, facilitating single-cell investigations, and catalyzing the creation of highly efficient customized biosensors. Nevertheless, hurdles persist, particularly concerning the increase in impedance and Johnson noise as electrode dimensions decrease. The development of nanoelectrodes stands at the cusp of a transformation with the rise in novel materials such as conductive polymers and hybrid organic–inorganic nanomaterials. They maintain mechanical robustness and electrical charge transfer capabilities even after miniaturization. This marks a pivotal shift from the classical metallic materials previously used in neural electrode fabrication. Researchers have turned to hybrid nanocomposites to enhance electrode performance without compromising structural integrity or operational lifespan [118,119,120,121]. These composites leverage two charge storage mechanisms concurrently. Hybrid electrodes are formed by embedding graphitic carbons within pseudocapacitive materials like conducting polymers and metal oxides [122,123,124]. These hybrids consistently outperform traditional electrodes, particularly in supercapacitor applications. Graphene and carbon nanotubes have emerged as leading candidates due to their atomically thin carbon structure, resulting in an extensive specific surface area, superior electrical conductivity, and impressive mechanical properties [125,126,127]. For instance, laser-scribed graphene has demonstrated a specific capacitance of approximately 202 F/g [128]. Meanwhile, bioinspired solvated graphene-based supercapacitors have exhibited an even higher capacitance of about 215 F/g. Moreover, chemically modified graphene has displayed a commendable capacitance of roughly 135 F/g [129]. Conducting polymers like polypyrrole (PPy) have undergone extensive study in pseudocapacitive materials. PPy exhibits superior redox performance, characterized by its cost-effectiveness, environmental stability, and suitability for large-scale processes. These properties significantly elevate overall performance when integrated into a hybrid graphene electrode, particularly in neural interfacing applications. This integrated approach enables more efficient and precise recording, stimulation, and interaction with neural tissue [130,131].

Kim, Gook Hwa et al. [132] have innovatively designed transparent electrodes employing a combination of graphene and vertically aligned carbon nanotubes (VACNT) for extracellular recording of spontaneous action potentials in primary cortex neurons of Sprague-Dawley rats. The graphene component fulfills a dual role: it establishes contact with the VACNTs and allows for the visual monitoring of cell viability. The hybrid electrodes display impressive performance, presenting significant peak-to-peak signal amplitudes (1600 μV) alongside minimal noise levels. This exceptional performance is credited to the close integration between the cells and the contoured carbon nanotubes (CNTs). Introducing transparent graphene vertically aligned carbon nanotube hybrid (TGVH) electrodes revolutionized the field by enabling optical cell monitoring alongside simultaneous extracellular signal recording. Recording spontaneous action potentials from cortical neurons through TGVH electrodes exhibited remarkably high signal amplitudes and signal-to-noise ratios (SNR). This heightened performance is a distinctive feature of carbon nanotubes (CNTs), owing to their porous network, surface properties promoting cell adhesion and proliferation, and impressive electrical conductivity. The vertically aligned carbon nanotubes (VACNTs) within the TGVH electrodes possess a protruded structure ideally suited for cellular interfacing (see Figure 8).

The study [107] introduced a novel hybrid neural probe design that integrates graphene, ZnO nanowires, and a conducting polymer. This architecture was engineered for flexibility and optimized low impedance performance. This was achieved through a hybrid structure combining Au and graphene, ensuring flexibility and conductivity. Employing ZnO nanowires to enhance the surface area significantly reduced impedance values, consequently improving the signal-to-noise ratio (SNR). Furthermore, applying a poly [3,4-ethylenedioxythiophene] (PEDOT) coating enhanced electrical properties and biocompatibility. In vivo recordings demonstrated the probe’s capability to detect more precise neural signals.

### 4.4. Biocompatibility

Biocompatibility, an essential consideration for implantable devices, integrates biological, chemical, and physical properties. Key objectives involve avoiding toxic or immunologic reactions, preventing harm to enzymes, cells, or tissues, and minimizing compression-related issues in adjacent tissues, emphasizing the need for robust biocompatibility in implantable technologies [133]. This approach aims to seamlessly integrate biomedical devices with living systems, ensuring longevity, functionality, and safety in diverse medical applications. The biocompatibility of implanted recording electrodes depends on various factors, including electrode materials, device geometry, and ambient surroundings. Biocompatibility, in the material context, is defined as the “ability of a material to elicit an appropriate host response in a specific application” [134]. Ideal biomaterials for neural recording implants should demonstrate in vivo non-cytotoxicity, releasing minimal substances at low, non-toxic concentrations. The desired outcome includes minimal glial encapsulation and a mild foreign body reaction, avoiding necrosis or implant rejection [135,136].

Critical assessments of material and device biocompatibility involve various tests, such as cytotoxicity, acute/chronic systemic toxicity, sub-acute/sub-chronic toxicity, sensitization, irritation, genotoxicity, hemocompatibility, toxicokinetic studies, and immunotoxicology [137]. The International Organization for Standardization (ISO) [138] establishes thorough test and evaluation protocols, considering potential variations in a material’s response across diverse biological environments. This approach considers factors like body contact types, contact time, intended use environments (in vitro, ex vivo, or in vivo), and test methodologies, as outlined by Hanson et al. [139]. Rigorous evaluations are essential for ensuring the compatibility and safety of neural recording implants across various physiological contexts.

Electrodes constructed from graphene have garnered substantial attention in monitoring neural activity. A crucial consideration entails the comprehensive evaluation of graphene and its derived materials in relation to human well-being, encompassing factors such as compatibility with biological systems, potential harm, and any environmental risks, particularly in situations involving incorporation with human skin or implantation. While studies on graphene-based nanomaterials (GBNs) abound, a discernible gap exists in systematic research regarding their effects on human health and the environment [140]. Safety evaluations are paramount in novel material development [141]. In the research literature, “graphene” broadly encompasses various GBNs, including GO and rGO [142,143]. Given the absence of standardized descriptions, key parameters for classification include layer count, average lateral size, and the carbon-to-oxygen (C/O) ratio, especially when considering various synthesis methods [142,144].

The physicochemical properties, including dosage, purity, shape, layers, surface chemistry, lateral size, and thickness, play a significant role in determining the toxicity of GBNs. These elements impact biodistribution, transference to secondary organs, aggregation, deterioration, and elimination [140,143]. Following exposure to neural cells or biomolecules, GBN properties and biological behavior dynamically shift, potentially leading to degradation or biotransformation. Moreover, these characteristics may evolve in different biological milieus over time, emphasizing the significance of in situ assessments for prospective applications. The materials employed in implantable electrodes must exhibit exceptional biocompatibility and minimal toxicity in human interaction. Hence, the evaluation of graphene-based implant safety holds paramount importance.

## 5. Unveiling Neural Activities with GFETs for In Vivo/In Vitro Applications

The field of graphene bioelectronics holds great promise for research, offering highly adaptable fabrication methods and biocompatibility for interfacing with complex biological entities such as brain tissues [145,146]. Graphene is considered an ideal candidate for fabricating field-effect transistor (FET) arrays because of its remarkable properties and capability of maintaining stable direct contact with the cells to record and amplify the neuronal activity signals. In this context, GFET arrays offer a promising approach for neural interfacing and real-time monitoring of both intra and extracellular activities [147,148]. Experimental studies have impressively showcased the seamless integration of graphene into neural networks. This integration has been characterized by the absence of cell disruption and the preservation of neuronal signals, all achieved without any observed adverse effects [19,149].

### 5.1. Neuronal Activity Recording Using GFET Devices: In Vitro

The first study of GFET was reported by Cohen-karni et al. in 2010, demonstrating signal detection from electrogenic cells via mechanically exfoliating single-layer graphene devices [68]. In practice, they successfully recorded the conductance signal of GFETs from spontaneously contracting embryonic chicken cardiomyocytes, achieving a noteworthy signal-to-noise ratio (SNR) > 4 when compared to similar planar devices. The amplitude of this conductance signal was modulated simply by adjusting the water-gate potential of the GFET device, underscoring a robust graphene–cell interface. Furthermore, by manipulating this water-gate potential, they demonstrated both n-type and p-type recording capability. This accomplishment in integrating a micro-scaled GFET device into living electrogenic cells has propelled further theoretical and experimental investigations, paving the way for the development of next-generation wearable microdevices tailored for recording neuronal activities [83].

Employing GFET arrays seamlessly integrated with cell membranes makes measuring action potentials (APs) from electrogenic cells viable. In a study by Hess et al., a groundbreaking array of solution-gated GFETs (SG-GFET) was pioneered, utilizing extensive graphene layers cultivated through chemical vapor deposition (CVD) on copper foil. This innovative technology effectively captured the action potentials of HL-1 cells, which exhibit characteristics akin to cardiomyocytes [150]. The HL-1 cells were meticulously cultured, forming a densely packed layer over the transistor array, showing robust and healthy growth. Differential interference contrast (DIC) imaging was employed to record cell signals. Recurring spikes corresponding to action potentials’ propagation across the cell layers were observed. The study documented a signal propagation speed spanning from 12 to 28 µm·s^−1^ and a root mean square (RMS) noise level measuring 50 µV. It is worth highlighting that by averaging five successive spikes, the signal-to-noise ratio (SNR) experienced a notable enhancement, reaching a value of 70. This observation underscores the potential for significantly improved signal clarity.

Kireev and colleagues pioneered the creation of a chemical vapor deposition (CVD)-grown graphene multielectrode array (GMEA) device specifically designed for conducting in vitro recordings of spontaneous neuronal spiking–bursting activity [146]. They produced an array measuring 1.4 mm × 1.4 mm, featuring 64 electrodes per chip. Rat embryonic cortical neurons were cultivated at 800 cells per mm_2_ density, as depicted in Figure 9a. By the 21st to 25th day, the culture had matured sufficiently to exhibit spontaneous electrical activity across the neural network. A single GMEA device detected eight channels, registering a typical burst every 5–15 s. The recorded bursting amplitude spikes reached up to 800 µV, and for most of the graphene channels, we observed non-bursting action potentials with amplitudes ranging from 50 to 150 μV. In addition, the studies demonstrated favorable signal-to-noise ratios (SNRs) of 45 ± 22 for HL-1 (cardiac-like cells) and 48 ± 26 for cortical neuronal networks, respectively (refer to Figure 9b). In another work, they fabricated CVD-grown SG-GFET arrays and cultured cortical neurons over the chip with 1500 cells per mm_2_ density for 14 days [151]. The cultured neurons generated spontaneous APs upon maturing, propagating through the neural network. The average amplitude was around 630 µV after recording 77 Aps (refer to Figure 9c,d). Also, the GFET arrays exhibited a high signal amplitude of approximately 200 µV, and SNR was above 3. Moreover, they designed feedline follower passivation to ensure a better interface between the gate electrode and neurons.

Veliev et al. [152] introduced a CVD-grown GFET array on diverse, flexible substrates, showcasing high sensitivity and minimal noise levels. This array was utilized to demonstrate the in vitro detection of spontaneous activity in hippocampal neurons. They cultured the hippocampal neurons on top of the GFET arrays in an 8 mm wide PDMS microfluidic chamber for 21 days to allow the neural network to reach electrical maturity. Subsequently, they administered a synthetic polymer (poly-L-lysin) coating onto the GFET arrays to facilitate bonding between the neuron membranes and the graphene channel. Upon interaction of the live neurons with the channel surface, they observed a 0.2 V positive shift in the charge neutrality point, signifying a reduction in graphene channel conductance. The positive shift resulted from the influence of the negative resting membrane potential. Furthermore, the authors engineered GFET arrays with varying W/L ratios, employing a high-quality CVD-grown monolayer graphene. This enhancement aimed to boost the transconductance and the quality of recording ion channel activity within hippocampal neural networks [153]. Hippocampal neurons were cultured on GFET arrays for 19–21 days, with a density of 0.5 × 105 cells/cm^2^, allowing them to reach electrical maturity. Following the methodology employed in their prior study, the GFET arrays underwent a coating process involving a synthetic poly-L-lysine polymer. This coating was applied to enhance cell adherence and promote outgrowth. In the recordings of neurons on functionalized GFETs, the conductance of the graphene stripe was determined by measuring the drain–source current. This was achieved while maintaining a constant bias voltage and liquid-gate potential. The neuron-induced signal was three times larger than the intrinsic electronic noise of the transistor. It was also observed that the signal amplitude of the drain–source current varied from 10 to 100 nA for more extensive and smaller devices, respectively, indicating a clear dependence on device size.

Kalmykov and colleagues introduced an organ-on-electronic chip featuring a 3D self-rolled biosensor array (3D-SR-BA) to monitor cell–cell communications within cardiac spheroids (see Figure 9e) [154]. This groundbreaking platform incorporates graphene microelectrodes and GFET arrays strategically arranged on a planar surface supported by a pre-stressed metal/polymer. This support structure allows controlled transformation into a 3D geometry upon release. The 3D-SR-BA device offers a high degree of customization, allowing for the configuration of electrodes and curvature to be tailored for specific applications (refer to Figure 9f). The researchers employed a novel approach for recording electrophysiological signals in stem cell-derived engineered cardiac spheroids. Direct contact between the biosensing device and the spheroid significantly improved the biosensor–cell interface. Twelve microelectrode biosensors were utilized to capture the field potentials (FPs) originating from the encapsulated cardiac spheroid, indicating a beating rate of 19 beats per minute. Subsequently, a Ca^2+^ indicator was introduced to the cardiac spheroids, and the observed Ca^2+^ spike frequency closely correlated with the FP spike frequency (see Figure 9g).

**Figure 9 sensors-23-09911-f009:**
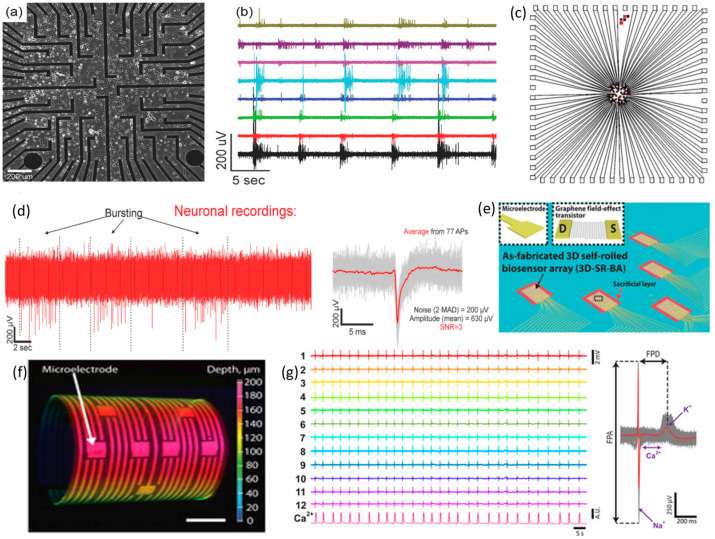
In vitro neural recording by graphene transistor arrays. (**a**) Microscopic image of a dense neuronal network cultured over a GMEA array; (**b**) a time-series recording of spiking–bursting activity propagating through different network channels [reprinted with permission from [146]]; (**c**) the design layout of 32 arrays in GFET chip for in vitro recording of neuronal signals; (**d**) time track recording of an intrinsic neuronal bursting activity displays the alternative burst periods at high frequency and spikes at low frequency and the average AP (red) obtained from 77 individual APs (grey) [reprinted with permission from [151]]; (**e**) a 3D self-rolled biosensor array fabricated on a sacrificial layer; insets D and S are the drain and source of GFET, respectively; (**f**) a 3D confocal microscopic image of this biosensor array, scale bar 50 µm; (**g**) recording of the FPs where Ca^2+^ fluorescence intensity was continuously recorded as a function of time and average FP peak (red) calculated from 100 peaks (gray) [reprinted with the permission from [154]].

### 5.2. Neuronal Activity Recording Using GFET Devices: In Vivo

GFET devices have made substantial contributions to monitoring brain activities in vivo. Hebert et al. fabricated a CVD-grown flexible SG-GFET array for micro-electrocorticography (µ-ECoG) recording of rat brains (refer to Figure 10a) [155]. They employed two arrays of transistors, one sized at 80 × 30 μm^2^ and the other at 100 × 50 μm^2^, for recording neural activity from the cortex surface. Maintaining a constant rain–source voltage of 100 mV, they kept the gate bias between −0.3 and 0.6 V. The transconductance averaged at 2.4 mS·V^−1^, demonstrating a field-effect mobility of 863 cm^2^·V^−1^ s^−1^, indicating remarkable homogeneity in the SG-GFET array within a saline solution. Moreover, the transistor demonstrated the capability to function at gate frequencies surpassing 10 KHz, accompanied by a mean equivalent gate noise of 21 µV. The researchers employed an array of transistors measuring 80 × 30 μm^2^ to document the synchronized activity within the cerebral cortex of WAG rats (see Figure 10b). The outcome of the frequency analysis unveiled oscillations in the range from 3 to 4 Hz, indicative of synchronous activities.

Yang and colleagues pioneered an innovative approach in fabricating electrocorticography (ECoG) probes for in vivo recording of rat brain epileptic activity, utilizing highly flexible and crumpled graphene transistor technology [156] (refer to Figure 10c). Utilizing a thin, chemical vapor deposition (CVD)-grown film of porous carbon nanotubes (CNTs), source–drain electrodes were structured and subsequently transferred onto a copper substrate. These intricately folded transistors were positioned on the cortical surface to capture the ECoG activity (see Figure 10d). Subsequently, sodium penicillin G was administered to provoke epilepsy in the rat brain, and real-time recordings of the induced epileptic activity were acquired (see Figure 10e). Notably, the Basel, latent period and epileptiform activity exhibited distinct characteristics. The latent period persisted for a few minutes, during which delta waves were subdued. Population spikes emerged from the synchronized firing of neuronal clusters throughout the epileptiform activity phase. This phase lasted for approximately 3 h, after which the spike activity gradually reverted to baseline.

Blaschke et al. reported the fabrication of highly flexible 16 SG-GFET arrays on polyimide substrates for in vivo recording of pre-epileptic activity [157]. They conducted measurements of the local field potential (LFP) within the anesthetized rat brain before implanting the arrays onto the surface of the cerebral cortex. The drain–source current was compared to the gate voltage while keeping a constant drain–source voltage for the in vivo evaluation of the transistor arrays. The pre-epileptic activity was instigated by locally injecting bicuculline for the neuronal recordings in rat brains. The graphene transistor arrays outperformed the advanced Pt electrodes of different sizes as they recorded significantly more significant spikes. The signal-to-noise ratio (SNR) reached an impressive value of 72, surpassing state-of-the-art Pt electrodes, demonstrating its potential. Moreover, this SG-GFET can operate at zero gate bias, showcasing excellent prospects for in vivo recording of neuronal activities.

Masvidal-Codina and colleagues pioneered the creation of flexible epicortical and intracortical SG-GFET arrays designed for in vivo recording of infra-low signals, particularly at sub 0.1 Hz frequencies. These arrays were also utilized for mapping cortical spreading depression (CSD) in rat brains [158]. Zero insertion force connectors made the interface between the recording electronics and the 12 m GFET arrays easier. Plotting the drain–source current against the gate voltage while holding the drain–source voltage constant was used to characterize the transistors. The charge neutrality point was found to have a small, 243 mV centered dispersion. Electrophysiological signals were captured over a wide bandwidth using the current source density (CSD) technique. Two craniotomies were performed on Wistar rats under isoflurane anesthesia, targeting the left hemisphere. The primary somatosensory cortex received a more extensive craniotomy, while a smaller one was executed in the frontal cortex. Induction of CSD was achieved by injecting a five mM KCL solution. Simultaneous recordings were acquired from two distinct frequency bands: a low-pass filtered range (LPF, 0–0.16 Hz) and a high-pass filtered range (BPF, 0.16 Hz to 10 kHz), with adjustable gains for each. The LPF signal revealed an extended CSD event, whereas the BPF signal, corresponding to the local field potential, indicated the cessation of CSD activity. Additionally, the authors utilized 4 × 4 epicortical SG-GFET arrays to map the propagation of CSD. These mapping results were subsequently compared to recordings obtained with high-pass filtering. The CSD events exhibited a duration of approximately 47 ± 8 s, with a propagation speed measured at 8 ± 1 mm/minute.

Garcia-Cortadella et al. introduced an innovative approach employing frequency division multiplexing (FDM) of SG-GFET for in vivo recording of neural activities [159]. This approach significantly minimized on-site switching, leading to a substantial technological simplification compared to time-division multiplexing (TDM), as shown in Figure 10f. The GFET sensor arrays effectively detected amplitude-modulated (AM) neural signals, which were then transmitted through a common communication channel. These sensor arrays were structured in an addressable column/row matrix configuration, facilitating the recording of wide-band neural activity on the surface of the rat’s brain. The electrical signals from the cortex of a Long Evans rat were recorded using a 4 × 8 frequency division multiplexing (FDM) graphene neural probe. The gate–source voltage was fine-tuned to optimize transconductance. The probe was successfully implanted in the primary visual cortex V1 (lower left), yielding a response with a peak amplitude of 250 μV and a delay of 50 ms, along with reduced crosstalk (see Figure 10g). Suppressed spontaneous current source density (CSD) activities also displayed a notable infra-low signal drift over 70 s under cortical anesthesia. In summary, these innovative FDM graphene neural probes exhibit high proficiency in monitoring wide-band oscillatory dynamics within the brain.

The author also pioneered the creation of flexible 64-channel SG-GFET sensor arrays, each measuring 100 × 100 μm^2^, enabling the wireless recording of epicortical brain dynamics in rats across a broad frequency range (see Figure 10h) [160]. The neural probes were created using a wafer-scale process with graphene grown through chemical vapor deposition (CVD), ensuring excellent uniformity and high production yield. We seamlessly integrated a specialized wireless head stage into the recording system to showcase the significance of signal amplification and digitization. Incorporating a two-stage trans-impedance amplifier effectively minimized quantization noise originating from the head stage and mitigated substantial DC offsets. Subsequently, the current flowing between the drain and source was converted to voltage within the amplification stages. This was followed by applying a high-pass filter to eliminate any remaining DC offset. We conducted a 24 h monitoring of naturally behaving rats, during which we utilized a motion capture (Mocap) system to track their 3D movements—the amalgamation of these data allowed for the classification of both brain state and behavioral state. Our stability test on the graphene sensors, conducted over four weeks, exhibited a consistently stable frequency response within the local field potential (LFP) frequency band. Also, a biocompatibility test using behavioral and histological indicators over 12 weeks showed an acute foreign body response comparable to platinum-based implants.

Bonaccini Calia and colleagues developed implantable depth neural probes utilizing flexible graphene microtransistors grown through chemical vapor deposition (CVD). These probes, known as GDNPs (graphene-based depth neural probes), enable synchronous in vivo recording of high-frequency neuronal dynamics with exceptional spatial resolution, particularly during seizures in a rat brain [161]. This study exposed graphene and reference gate electrodes to neural tissues. The graphene dual-node probe (GDNP) was created on a polyimide substrate with a thickness of 10 μm, featuring 14 linear transistor arrays (measuring 60 × 60 μm^2^) set at a pitch of 100 μm. A two-level metallization approach was employed to optimize sensing performance and minimize track resistance. Insertion of the GDNP into the mouse cortex was facilitated using silk fibroin (SF). Following probe insertion in the right hemisphere visual cortex (V1), which ensured proper contact with the hippocampus tissue, electrophysiological signals were captured in awake, head-fixed mice. Baseline recordings further confirmed successful probe placement in the hippocampus, demonstrating theta activity. Subsequently, the GDNP recorded approximately seven seizures within a 60 min post-drug injection period. Over ten weeks, the authors obtained chronic full-bandwidth recordings from the implanted GDNP in the somatosensory cortex of the right hemisphere using a rat model of absence epilepsy. Additionally, the GDNP demonstrated the capability to capture high-fidelity spontaneous spike–wave discharges and associated inflow oscillations. In Table 2, a comparison is presented for different graphene field-effect transistors (GFETs) used in recording neuronal activities.

**Figure 10 sensors-23-09911-f010:**
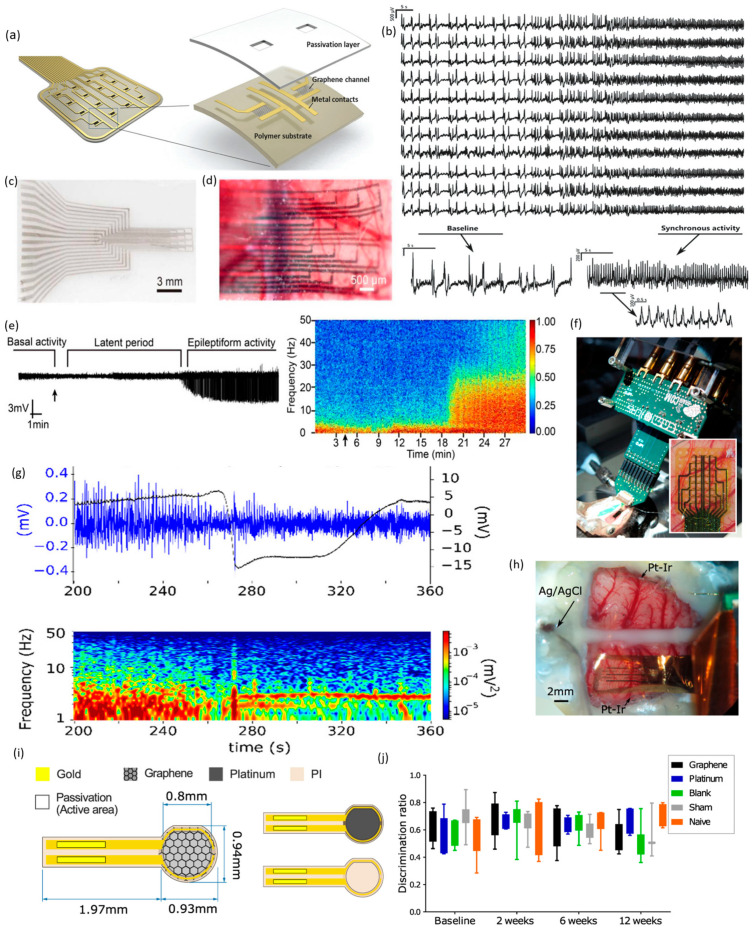
GFET devices were utilized for in vivo electrophysiological mapping. Panel (**a**) displays a pictorial representation of the SG-GFET array featuring various components. In panel (**b**), a synchronous LFP (local field potential) was recorded from the cerebral cortex of WAG rats, exhibiting a frequency range of 3–4 Hz [reprinted with the permission from [155]]. (**c**,**d**) Optical photographs of the crumpled GFET arrays taken before and after positioning them over the left cortical surface of the rat’s brain. (**e**) Live monitoring of induced epilepsy activities featuring three distinct phases; the penicillin injection time is marked by the black arrow [reprinted with the permission from [162]]. (**f**) The experimental configuration depicting the interfacing of the SG-GFET array with the brain, coupled with a custom-built front-end amplifier. (**g**) Captured recording illustrating a CSD propagating front using a single SG-GFET. Activity within the 1–50 Hz frequency range is depicted in blue (left axis), while wide-band activity (0.001−50 Hz) is represented in black (right axis), alongside the corresponding spectrogram within the 1−50 Hz band. [Reprinted with the permission from [159]]. (**h**) Array of SG-GFET positioned on the rat cortex. (**i**) Illustration outlining the SG-GFET prototype for conducting in vivo biocompatibility assessments. (**j**) The evaluation of discrimination ratio was conducted using the novel object recognition test on various days following the implantation. [Reprinted with the permission from [160]].

## 6. Equivalent Circuit Modeling and Addressing Acute Inflammation

Implanted electrodes play a crucial role in neural recording by detecting alterations in extracellular potential resulting from ion exchange in nearby regions, allowing for the capture of low-frequency local field potentials (LFPs) (<~350 Hz). They can even capture individual neurons’ action potentials (APs) (~kHz) in specific scenarios. Impedance is a pivotal characteristic of implantable electrodes, representing the resistance to current flow between the electrode and neural interface. It estimates the capacity of the electrode for recording the compulsive and functional neural signs or stimulating them in a closed-loop application. An equivalent circuit model is necessary to understand the neural tissue interface’s functionality, which can often be understood, modeled, and optimized for BCI applications (refer to Figure 11). Here:

V_e_ = a low-impedance voltage source and signaling in the neuron.

R_spread_ (or R_media_) = implantable electrode’s impedance.

R_e_ = implantable electrode’s leakage resistance.

C_e_ = electrode–tissue interface’s capacitance.

R_s_ = resistance of the external electronic.

A smaller impedance is generally anticipated and favorable for rapidly recording signal (V_e_). Like the recording, stimulation of implantable electrodes can be improved with a low resistance of the electrode–tissue interface and results in a significant charge injection toward the neurons. Therefore, low impedance is necessary for interface location for seamless neural recording/stimulation [163,164,165].

The defects of the material, inadequate encapsulation, and perhaps the accidental mechanical pressures produce delamination and fracturing of the recording sites of the implantable electrodes [166]. The failure of encapsulation may occur between the first few days or a month after the neural implantation, uncovering the metal’s interconnection. The damage due to the insulation creates a further barrier of resistance and capacitance for the current flow. The incident adds VNT noise to the neural signal during signal recording [167,168]. In addition, the amplitude of the recorded neural signal may experience attenuation due to the presence of low-impedance pathways in the vicinity of the recording sites. Similarly, these equivalent shunt pathways might divert the respective current and activate non-target neurons, diminishing the electrode’s overall stimulation efficacy. The corrosion introduces a dual negative impact because of the worsening of the chemical material used in the electrode. First, it ruins the metal’s interconnection’s conductive properties (increasing R_s_ and shrinking C_e_) [169]. Secondly, the interface may generate harmful toxic ingredients, which can enhance the immune response or lead to the death of neural cells. Therefore, it is crucial to carefully select materials and prioritize systemization for an implantable electrode that is chemically durable and adequately insulated.

## 7. Microelectrode Array for Recording/Stimulation Applications

Quantum capacitance, a phenomenon resulting from the low density of states in graphene at the Dirac point, affects the impedance of graphene electrodes [170]. In 2010, Tzahi Cohen-Karni and colleagues introduced the first graphene-based transistor for electrical recordings and cellular interfaces [68]. A flexible microelectrode based on graphene was created in 2013 by Chang-Hsiao Chen et al. for recording brain and cardiac functions. Mild steam plasma (SP) treatment was used to add C–OH, C=O, and OH–C=O bonds to the surface of the graphene layer to increase its hydrophilicity. Consequently, this treatment led to a reduction in graphene impedance (from 7216 to 5424 Ω/mm^2^) and an increase in signal-to-noise ratio (SNR) during electrochemical detection (from 20.30 ± 3.31 dB to 27.81 ± 4.03 dB), while maintaining a stable bias voltage without drift. This improvement in the treated graphene’s performance allowed for the precise separation of neural signals and facilitated the identification of their distinct shapes. Additionally, due to the elevated SNR of the treated graphene, the QSR complex, P wave amplitude, and T wave in the zebrafish heart’s ECG showed notable improvements. After the SP treatment, the graphene surfaces’ interfacial bonding energy was reinforced, attributed to stronger van der Waals forces (see Figure 12).

The transmittance of doped graphene exceeds that of ITO and ultrathin metals. This enabled the research team to conduct optical coherence tomography (OCT), optogenetic stimulation, and fluorescence imaging directly beneath the graphene electrode sites. The fact that these techniques remained error-free even when intense light was shone on an electrode location is noteworthy [88]. A separate research team in China fabricated biocompatible graphene microelectrode arrays (MEAs) and provided evidence of their exceptional long-term stability in aqueous solutions. [172]. Enhancing impedance can be achieved through an increase in geometric or surface areas. It is important to note that while electrode transparency may decrease with heightened surface roughness, this enhancement promotes recording action potentials.

At 1 kHz, the impedance exhibited a reduction of approximately 70%, decreasing from 2.9 ± 0.4 MΩ for the untreated graphene to 42 ± 2 kΩ for the PEDOT: PSS electrodeposited electrode. Conversely, transparency saw a decline with longer electrodeposition times. The optimal combination of impedance (166 ± 13 kΩ, showing an 18-fold reduction) with high transparency (84.4 ± 4%) was achieved through a 1 s PEDOT: PSS deposition. Cardiac field potential recordings conducted on graphene electrodes coated with PEDOT: PSS demonstrated peak-to-peak amplitudes reaching 3.8 mV and noise levels below 20 µV peak-to-peak. The coated electrode’s transparency made it easier to use fluorescence microscopy to specifically target the ganglion cells tagged with EYP during neuroimaging of the transgenic mouse retina [173]. In another study, researchers observed a significant reduction, approximately 100 times, in the impedance of the graphene electrode through the electrodepositing of Platinum nanoparticles (Pt NPs), effectively overcoming the quantum capacitance limit. Importantly, the transparency of the electrode remained unaffected by the presence of deposited Pt NPs. Additionally, in vivo experiments conducted on transgenic mouse models demonstrated the practical utility of this microelectrode array (MEA) in combining electrophysiology with optical imaging, a feat unattainable with opaque electrodes [174].

Efficient stimulation is essential for various electrode applications, including deep brain stimulation, spinal cord stimulation, cochlear and retinal implants, and cortical, intra-cortical, and peripheral nervous system applications. Achieving this requires a minimum cathodic current (CIC) level to depolarize the excitable membrane cells near the stimulating electrode. Typically, the electrode material should deliver a sufficient charge, ranging from 10 μC·cm^−2^ to mC·cm^−2^, within a pulse duration of 100 μs to 1 ms, contingent on the specific neural tissue being stimulated. The double-layer capacitance and its potential window when submerged in water determine the capacity for charge injection. The charge injection capacity (CIC) for chemical vapor deposition (CVD)-grown graphene ranges from 5 to 20 μF·cm^−2^ for a 1 V potential window in water. This value is slightly lower than conventional metals like Pt and Au commonly used for electrode fabrication. It also falls well below that of alternative novel materials like PEDOT–CNT or IrOx, as recent research [141] has suggested, for neural stimulation. The electrochemical properties of chemical vapor deposition (CVD)-grown graphene from various studies are outlined in Table 3.

In a study led by Heo et al., a graphene/Polyethylene Terephthalate (P.E.T.) electrode was used to modulate neural cell interactions in vitro through non-contact electrical stimulation. The electrode assembly was connected to a stimulator and placed in a culture dish. The study observed distinct cell-to-cell responses, categorized into waving, decoupling, coupling, and strengthened coupling. Graphene’s high transparency across different light spectra, from infrared to ultraviolet, enables the integration of electrophysiology with techniques like calcium imaging and optogenetics [67]. Another experiment by Dong-Wook Park et al. demonstrated that graphene-based neural electrodes allowed simultaneous imaging of neural responses to electrical stimulation, outperforming traditional materials. The charge density delivered by graphene electrodes fell within safe limits for tissue activation [177,178].

## 8. Future Scope

While graphene has already demonstrated its potential in creating implantable electrodes, there remains a vast untapped potential for its application in our chosen field. By combining graphene with other nanomaterials and polymers, we can modify its physiomechanical properties to enhance its performance. Specifically, organic polymers like PDMS, known for their exceptional flexibility, can be integrated with graphene to form composites crucial for developing highly flexible neural prototypes. While the electrical conductivity of these nanocomposites may be somewhat compromised compared to their pure counterparts, strategic optimization of the nanofillers’ percolation threshold within the composites holds the key to producing sensors equipped with high-quality electrodes. Fabricating electrodes on biocompatible, flexible substrates such as fabrics, requires a straightforward process and ensures prolonged durability. Thanks to graphene’s biocompatible properties and its effectiveness in neural recordings, these devices hold the potential for diverse applications in brain–computer interfacing. This potential encompasses the development of electrode arrays on commonly used substrates. To enable real-time applications, the electrodes used for neural recordings can be seamlessly integrated with signal-conditioning circuitries, reducing the need for laboratory environments.

Furthermore, by incorporating wireless communication protocols into the sensory units, data can be efficiently transmitted to monitoring units for further processing. This integration could include using radio-frequency identification (RFID) tags, enabling comprehensive monitoring of patients in innovative or specialized living arrangements. These graphene-based electrodes offer a promising avenue for detecting anomalies in their daily routines, providing invaluable insights for personalized care and intervention. From a structural point of view, the variations in the shape of electrodes and the corresponding working principle can help use graphene-based electrodes in multilayered structures. The varied forms of the prototypes can help users integrate the sensors with flexible circuit boards to support work in a broader range of neural-related biomedical applications. The large number of electrodes used for a complete sensing system is one major issue that needs to be resolved. Graphene to form different implantable electrodes on a singular substrate should be encouraged to reduce the system’s complexity and cost. This would reduce the simplification of the recording process and increase the graphene electrodes’ capability in sensitivity and efficiency.

## 9. Conclusions

In conclusion, graphene emerges as a paramount material in advancing neural electrode technology, owing to its exceptional optical transmission, electrical conductivity, chemical stability, and flexibility. This comprehensive review sheds light on the pivotal role of graphene and its derivatives in neural electrodes. The findings unequivocally establish that electrodes based on graphene significantly improve the signal-to-noise ratio in recording, resulting in superior outcomes for both stimulation and recording while minimizing artefacts in optical imaging. This leads to an enhanced spatial and temporal resolution in signal detection.

Furthermore, neural electrodes employing graphene-related materials effectively mitigate immune responses in brain tissue, augmenting the flexibility and durability of the electrodes. Compared to metal electrodes, graphene-related materials also demonstrate heightened sensitivity in recording neuronal signals, offering a cost-effective alternative. Integrating graphene and two-dimensional materials presents unprecedented opportunities in developing ideal neural electrodes. Addressing these challenges, from extending the optical response to developing bioactive electrodes, signifies a promising trajectory. Bridging the gap from animal models to clinical trials is imperative, and resolving the neurotoxicity associated with graphene-related materials remains a critical pursuit. Pursuing multifaceted graphene composite electrodes for diverse applications in neuroengineering and brain-related disorders represents a compelling and vital avenue for future research and development. The endeavors in this direction are poised to redefine the landscape of neural interfacing and contribute significantly to advancements in neuroscience and technology.

## Figures and Tables

**Figure 1 sensors-23-09911-f001:**
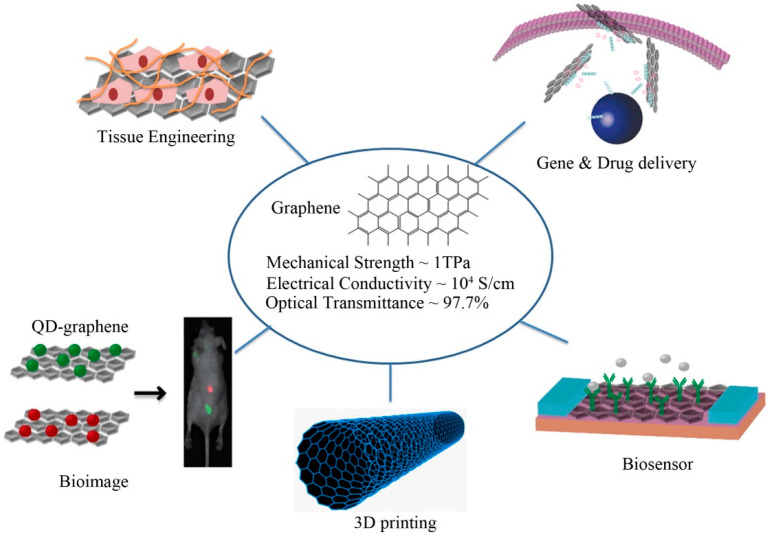
Extensive utilization of graphene-based materials in regenerative medicine and tissue engineering. [Reproduced with permission from [71]].

**Figure 2 sensors-23-09911-f002:**
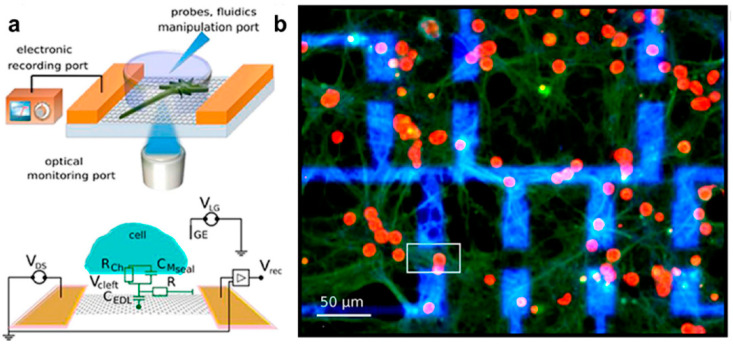
Graphene microelectrodes are utilized for in vitro recording of neural activity. (**a**) Illustration of the experimental arrangement featuring transparent graphene electrodes seamlessly integrated with an inverted microscope. (**b**) Images from fluorescence microscopy demonstrate well-established cultured neurons flourishing on the surface of graphene field-effect transistors. [Adapted from [58]. Copyright (2017), with permission from Frontiers].

**Figure 3 sensors-23-09911-f003:**
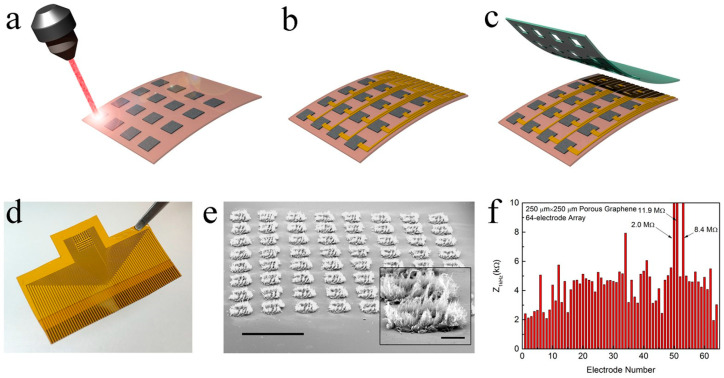
The porous graphene electrode array is being made. The diagrams below show the steps involved in making the porous graphene electrode array: a photograph showing the finished 64-electrode array is displayed, a tilted scanning electron microscopy (SEM) image of the 64-spot porous graphene array is shown, and impedance measurements of the 64 electrodes were carried out at 1 kHz. (**a**) Using laser pyrolysis to pattern the graphene; (**b**) establishing metal interconnects; (**c**) applying SU-8 encapsulation; (**d**) Image capturing a created 64-electrode array; (**e**) Tilted scanning electron microscopy (SEM) depiction of a 64-spot array composed of porous graphene. The inset showcases an SEM view of an individual spot; and (**f**) Evaluation of impedance for all 64 electrodes at 1 kHz. [Reprinted with the permission of [105]].

**Figure 4 sensors-23-09911-f004:**
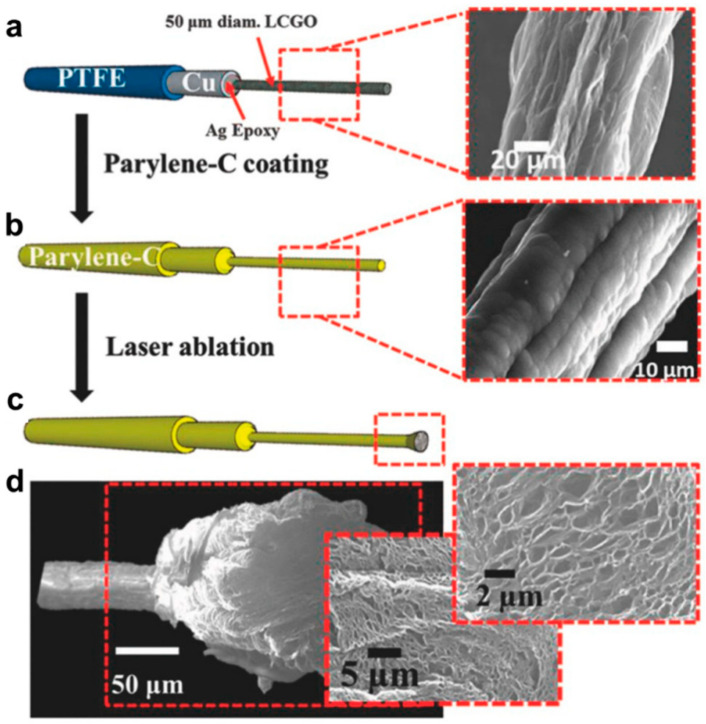
Manufacturing and visualizing LCGO brush electrodes. (**a**) The electrodes are connected to copper wires and insulated with polytetrafluoroethylene, and possess an approximate diameter of 1 mm. This bonding is achieved using a conductive epoxy containing silver. (**b**) After this bonding, a layer of Parylene C is applied as a protective coating. (**c**) A laser operating at 250 mW is utilized for ablation. This step opens up the end of the electrode, resulting in the formation of a distinctive ‘brush’ electrode. (**d**) The application of laser treatment leads to the formation of an amorphous electrode, characterized by an exceptionally high degree of surface irregularities and porosity [reprinted with the permission from [106]].

**Figure 5 sensors-23-09911-f005:**
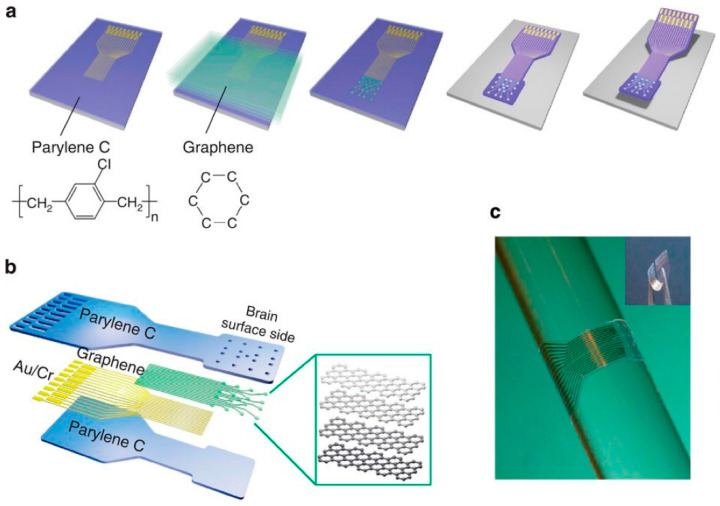
It illustrates the clear micro-ECoG device, highlighting key fabrication steps. (**a**) Initial metal patterning on a Parylene C-coated silicon wafer substrate for traces and pads. (**b**) Sequential stacking of four graphene monolayers. (**c**) Precise graphene patterning to form electrode locations. [Reprinted with the permission from [88]].

**Figure 6 sensors-23-09911-f006:**
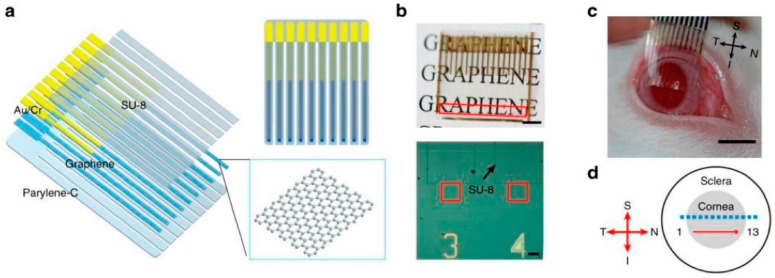
This figure illustrates a multielectrode ERG recording employing a soft and transparent graphene electrode array. The construction of the array involves layered structures (**a**). The top section displays the array’s optical transparency when positioned over printed paper, with recording sites arranged linearly (**b**). The bottom part offers an optical microscopy view, emphasizing graphene electrode sites and traces, including an insulated electrode (**c**). A stripped graphene electrode array is also shown over a dilated rabbit eye (**d**). The schematic showcases the distribution of recording channels on the rabbit eye, from the temporal area to the nasal periphery (**d**). [This figure has been adapted from [117]].

**Figure 7 sensors-23-09911-f007:**
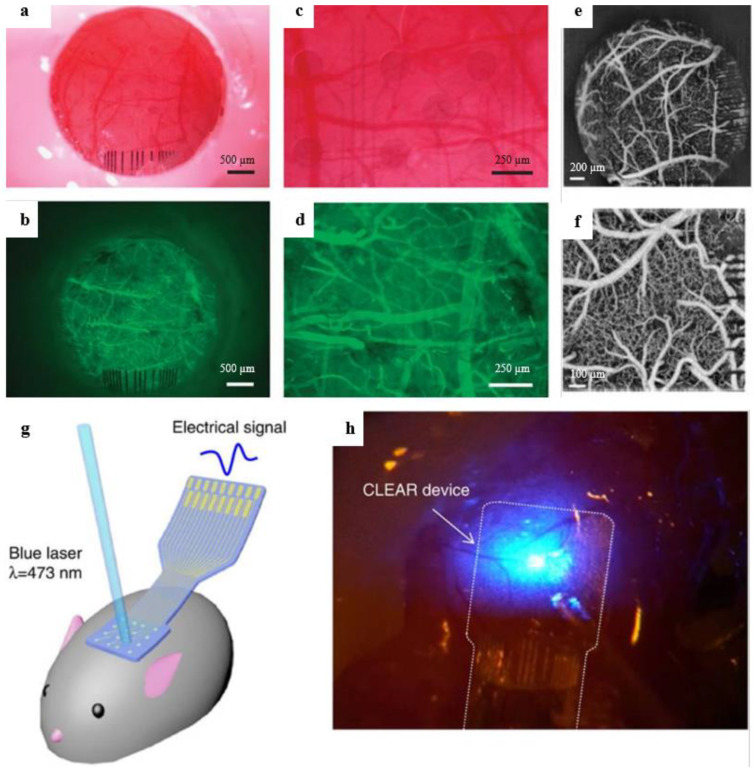
In vivo cortical vasculature images were captured using the CLEAR device. Panels (**a**,**c**) present the bright-field image of the graphene electrode on the cerebral cortex beneath a cranial window. Correspondingly, panels (**b**,**d**) showcase the fluorescence images of the same device as shown in (**a**,**c**). The cortical vasculature was visible through the graphene electrode in panels (**e**,**f**). A schematic illustrating optical stimulation by blue light with a 473 nm wavelength on the cerebral cortex is provided in panel (**g**), demonstrating its compatibility with a transparent graphene MEA. Lastly, panel (**h**) displays the recording of neural signals evoked by blue light via the transparent graphene MEA. (Reproduced with permission from [88]).

**Figure 8 sensors-23-09911-f008:**
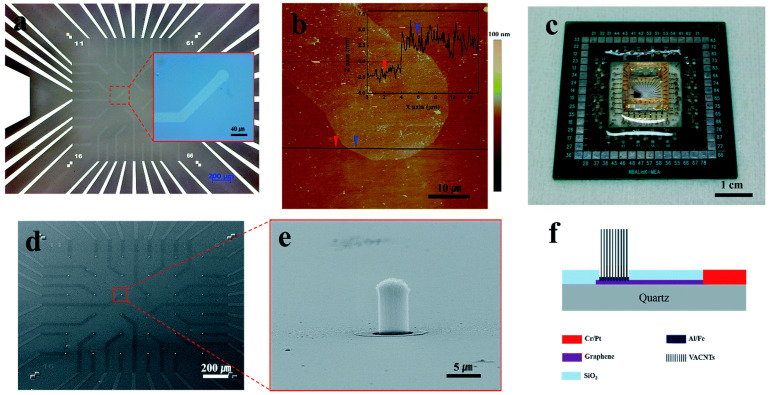
The neuronal signals were captured using a TGVH device. Panel (**a**) exhibits optical images of a custom-designed electrode array composed of patterned graphene. This array comprises 35 distinct graphene electrodes, each with 1 × 1 mm dimensions, accompanied by an internal ground electrode spanning 2.9 mm^2^, all positioned on a Cr/Pt base electrode. The inset provides a closer view of a single-channel graphene electrode. In panel (**b**), a topographical AFM image reveals a two-layer graphene electrode, with the inset indicating the thickness of the marked line. Panel (**c**) displays the finalized TGVH device. Panels (**d**,**e**) show FE-SEM images, respectively, of the multielectrode array constructed from vertically aligned carbon nanotubes (VACNT) in its original state and a single VACNT electrode. Finally, panel (**f**) presents a schematic representation of the TGVH device. [Reprinted with the permission of [132]].

**Figure 11 sensors-23-09911-f011:**
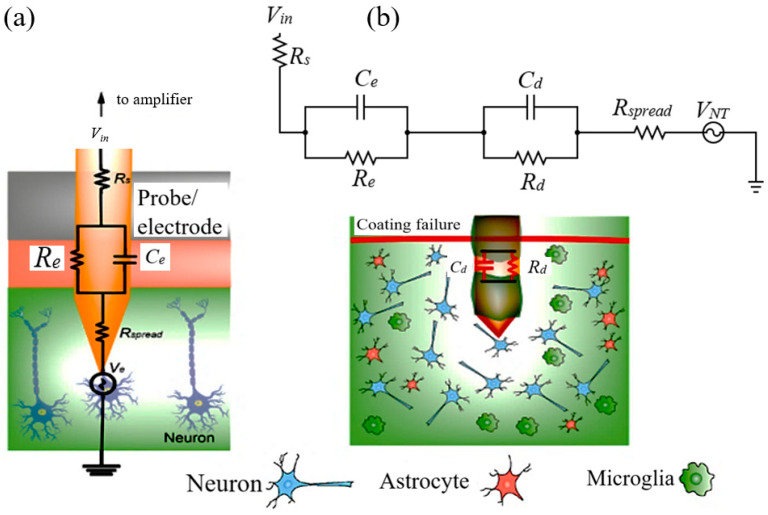
(**a**) This schematic illustrates the equivalent circuit representing the interface between the probe and neural tissue locations. Only the neural recording process is depicted to simplify the representation, with neurons acting as a voltage source (V_e_). However, it is important to note that an analogous neural stimulation circuit can be characterized as well. (**b**) Depicted here is the scenario of an implantable neural device failure, along with its corresponding equivalent circuit.

**Figure 12 sensors-23-09911-f012:**
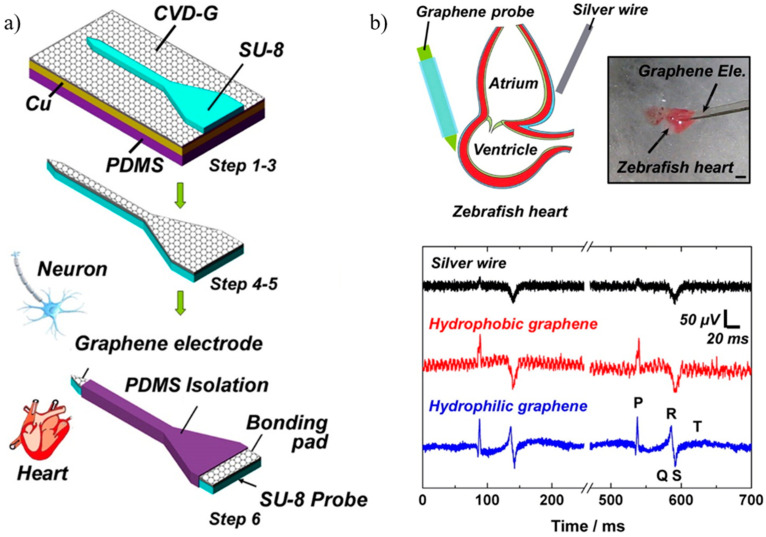
(**a**) Illustration depicting the process of graphene electrode fabrication. (**b**) Electrocardiogram (ECG) of a zebrafish heart. (Reprinted with the permission from [171]). The adhesion of graphene on the electrode has been enhanced, a crucial factor for ensuring long-term implantation. (Reproduced with permission from [171]).

**Table 1 sensors-23-09911-t001:** Comparative properties of flexible electrode materials.

Electrode Material	Electrical Property	Young’s Modulus	Transparency	Biocompatibility	Reference
PEDOT	High conductivity: 1200 S·cm^−1^	Flexible: 2.6 ± 1.4 GPa	Limited	Excellent	[89,90]
PT	Moderate conductivity: 10–100 S·cm^−1^	Stiff: 3 GPa	Limited	Moderate	[91]
PPY	Variable conductivity: 40–200 S·cm^−1^	Moderate flexibility: 430–800 MPa	Limited	Good	[92,93]
PANI	Low conductivity: 5 S·cm^−1^	Stiff: 2–4 GPa	Limited	Moderate	[94]
Graphene	Excellent conductivity: 243.5 ± 15.9 kΩ (~200 µm diameter)	Extremely flexible: ~1 TPa	High	Excellent	[88,95]
Carbon nanofiber (CNF)	Moderate conductivity: ~1 MΩ (2 cm length, 25.7 × 16.6 µm^2^)	Variable stiffness: 6–207 GPa	Limited	Good	[96,97]
Glassy carbon	Good conductivity: 11.0 ± 5.4 kΩ (300 µm diameter)	Stiff: 20 GPa	Limited	Good	[98]
Diamond	Moderate conductivity: ~207.9 kΩ (0.0079 mm^2^)	Very stiff: ~103 GPa	High	Excellent	[99]

**Table 2 sensors-23-09911-t002:** Comparison of different GFETs for recording neuronal activities.

Neural Interface	Signal Recording	Synthesis Technique	Substrate(s)	SNR	Fabrication	FET Active Area	Application	Reference
Graphene and Si nanowire FETs interfaced with embryonic chicken cardiomyocyte cells.	In Vitro	ME	SiO_2_/Si	>4	E-beam lithography	20.8 μm × 9.8 μm, 2.4 μm × 3.4 μm	Recording extracellular signals.	[68]
Cardiomyocyte-like HL-1 cells seeded over solution-gated GFET arrays.	In Vitro	CVD	sapphire	70	Photolithography	10 μm × 20 μm	Recording action potentials of cardiomyocyte-like HL-1 cells.	[150]
Cardiomyocyte- like cell line HL-1 cultured over the encapsulated GMEA.	In Vitro	CVD	Borofloat glass and SiO_2_/Si	45 ± 22 for cardiac and 48 ± 26 for neuronal bursting activity	E-beam lithography	1.4 mm × 1.4 mm	Cardio and neuronal extracellular recordings.	[146]
The primary cortical neurons and HL-1 cells seeded over solution-gated GFET arrays.	In Vitro	CVD	SiO_2_/Si, HfO_2_/Si, and polyimide/Si	>3	E-beam lithography	Different W/L: width—2, 5, 10, and 20 μm; length—3, 8, and 18 μm	Recordings of HL-1 cell line and cortical neurons.	[151]
Primary hippocampal neurons cultured over GFET arrays.	In Vitro	CVD	Si/SiO_2_, sapphire, glass coverslip, and polyimide	2.5	Photolithography	20 × 15 μm^2^	Detection of the spontaneous activity of hippocampal neurons.	[152]
Primary hippocampal neurons cultured over GFET arrays coated with poly-L-lysine.	In Vitro	CVD	sapphire, glass coverslip, and silicon on insulator	3	Photolithography	1000 × 250 μm^2^, 40 × 250 μm^2^, 40 × 50 μm^2^ and 20 × 10 μm^2^	Field-effect detection of ion channel activity within hippocampal neuronal networks.	[152]
3D self-rolled arrays of GFET interfaced with human cardiac spheroids.	In Vitro	LPCVD	SiO_2_/Si	6.6	Photolithography	Inner diameter of ~160 µm for single-turn	Recording cell–cell communications of cardiac spheroids.	[154]
Solution-gated GFET probes interfaced over rat cortex.	In Vivo	CVD	polyimide and SiO_2_/Si	-	Photolithography	80 × 30 μm^2^ and 100 × 50 μm^2^	Micro-electrocorticography (µ-ECoG) recording cortical activity.	[155]
Highly crumpled graphene transistor placed over the cortex.	In Vivo	CVD	SiO_2_/Si and elastomer	-	UV lithography	100 μm × 100 μm	Electrocorticography (ECoG) for recording brain epileptic activity.	[15]
Solution-gated GFET array interfaced over rat cortex.	In Vivo	CVD	polyimide	Up to 72	Photolithography	20 μm × 15 μm	Recording spontaneous slow waves, visually evoked, pre-epileptic activities.	[162]
Solution-gated GFET arrays placed in zero insertion force connectors and interfaced over the cortical surface.	In Vivo	CVD	polyimide	-	Photolithography	100 × 50 μm^2^	Mapping cortical spreading depression and infra-low brain activities.	[158]
Solution-gated GFET array placed on the right hemisphere in the brain surface.	In Vivo	CVD	SiO_2_/Si	-	Photolithography	50 μm × 50 μm	High-performance FDM for sensing wide-band neural activity.	[159]
Solution-gated GFET neural probes placed in zero insertion force connectors and interfaced on the right hemisphere in the pial surface.	In Vivo	CVD	polyimide	-	Photolithography	100 μm × 100 μm	Wireless mapping of the wide frequency band epicortical brain activity.	[160]
Graphene microtransistor-based depth neural probes implanted in the right hemisphere visual cortex.	In Vivo	CVD	polyimide	>1.26	Photolithography	60 × 60 μm^2^	Recording DC-shifts and high-frequency neuronal activity in awake rodents.	[161]

**Table 3 sensors-23-09911-t003:** Overview of the characteristics of graphene in recording/stimulation applications.

Application	Electrode Size (μm^2^)	Substrate	Impedance @ 1 kHz in KΩ	SNR	CSC μC/cm^2^	CIC μC/cm^2^	Tissue Type	Number of the Graphene Layers	Ref.
Neural and cardiac recording	13,500	SU-8	0.7	20			Heart tissue	1	[171]
Stimulation for treating neuronal disorders	7854 (D = 100) 17,671 (D = 150) 31,416 (D = 200)	Parylene C	286.4 ± 62.6 284.7 ± 125.0 215.7 ± 120.4		87.8	57.13	In vivo (Brain)	4	[115]
Improved neural interface for stimulation	310,000	Borosilicate glass	10		1248.7 ± 41.5			1	[175]
Neural interface for recording	314	Quartz glass	170	10				1	[172]
Neural imaging and optogenetics for neural interfacing	31,400	Parylene C	243				In vivo (Brain)	4	[88]
Neuroimaging and recording for high spatiotemporal resolution mapping of dynamic neuronal activity	2500	Polyimide	541	31			In vivo (Brain)	1	[59]
Electrochemical characterization for neuronal implants	7854 (D = 100)	Borosilicate glass	2.3 MΩ	35.8	910 ± 0.13	150 ± 0.05	Cortical rat neurons culturing	1	[176]

## Data Availability

Data are contained within the article.

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
