# Peer review of "Recent Advancements in Graphene-Based Implantable Electrodes for Neural Recording/Stimulation"

_sensors, 2023, doi:10.3390/s23249911_

Round 1
Reviewer 1 Report
Comments and Suggestions for Authors
This is a nice piece of work highlighting the potentials of graphene for umplantable electrodes. The number of self-citations is fine with 9 out of 155.
The tables at the end of the article omn GFETS and graphene recording/stimulation are very helpful to provide some overview to the reader, especially newcomers in the field. It would be wonderful if such a table would also exist for graphene inmidst of its material competitors.
The following improvements should be considered:
1) Sections 3 and 4 are lengthy and consist of too much rephrasing results but do little comparison and discussion.
2) The section on biocompatibility is too short and does not provide toxicological depth. Moreover the statement in lines 858 - 866 seems to be contradictive to lines 556 ff.
3) Overall, quantitative arguments in terms of physical parameters should be preferred rather than qualitative statements like "performs better / exceeds" and so on. (especially Sections 271 ff, 891 ff
4) The Utah Array or Michigan electrode should not be omitted.
5) References [20 - 22] are inappropriate.
6) Line 123 please explain CIC
7) Please explain the function of the electrodes rather than listing applications.
8) 228 needs the ITO comment provided in 446
9) section 6 is too short
Comments on the Quality of English Language
Line 92 "Enter" ???
Line 109 "is" ???
Line 461 Hybrrid
Reviewer 2 Report
Comments and Suggestions for Authors
The authors reported that “Recent Advancements in Graphene-Based Implantable Electrodes for Neural Recording/Stimulation”and quoted lots of related works. But there is still room for improvement in paper arrangement. This review could be accepted after modification as follows.
1. In introduction section, please make it clear that this review differs from published reviews.
2. “3. Potentiality of Graphene for Implantable Electrodes” may be blended into other parts.
3. The classification should be reprogrammed.
-4. Characteristics of Graphene-Based Implantable Electrodes
---4.1 Flexible Electrodes
---4.2 Transparent Electrodes
---4.3 Hybrid Graphene Electrode
---4.4 Biocompaitability
4. A more detailed outlook is needed.
5. Most sentences contain grammatical or spelling mistakes. Please the authors should check it carefully.
For example, “4.3 Hybrrid Graphene Electrode” should be changed to “4.3 Hybrid Graphene Electrode”
In Figure 8, the label should be adjusted to for more clearly understanding.
Comments on the Quality of English LanguageMost sentences contain grammatical or spelling mistakes. Please the authors should check it carefully.
